# Locomotor activity patterns of takin (*Budorcas taxicolor*) in a temperate mountain region

**Wenbo Yan**[1,2], **Zhigao Zeng**[2]*, **Huisheng Gong**[3], **Yan Duan**[1], **Leigang Zhao**[3], **Aliu Peng**[3]

**1** Shaanxi Key Laboratory of Bio-Resources, Shaanxi University of Technology, Hanzhong, Shaanxi, China, **2** Key Laboratory of Animal Ecology and Conservation Biology, Institute of Zoology, Chinese Academy of Sciences, Beijing, China, **3** Foping National Nature Reserve Administration, Foping, Shaanxi, China

* zengzhg@ioz.ac.cn

**Data Availability Statement:** All relevant data are within the manuscript and its Supporting Information files.

**Funding:** This research was supported by the National Natural Science Foundation of China

## Abstract

Understanding locomotor activity patterns would reveal key information about an animal's foraging strategy, energy budget and evolutionary adaptation. We studied the locomotor activity patterns of the takin (*Budorcas taxicolor*) in a temperate mountain region in China using GPS radio-collar technology from 1 July 2014 to 30 June 2015. Our research showed that takin had a bimodal crepuscular locomotor activity pattern, with an especially obvious movement peak at dusk. The takins showed significant seasonal differences in their movement rates, with the lowest movement rate in winter. The animals also showed sexual differences in their movement rates. In spring, the female movement rate was significantly higher than that of males during daytime, while during nighttime the movement rate of males was higher than that of females. The male movement rate was significantly higher than that of females in summer. The movement rate of the takins were correlated to microenvironment temperature and normalized difference vegetation index (NDVI) in each season. These findings suggest that takin could adjust locomotor activity levels adapt to reproductive requirements, temperature variation and forage variability.

## Introduction

Locomotor activity patterns of animals can be affected by various factors, such as ambient temperature and seasonal change [1–4], forage quantity and quality [2, 5], and sex and reproductive status [6–8]. Therefore, investigating locomotor activity patterns could reveal key information about an animal's foraging strategy, energy budgets and evolutionary adaptation [9].

In temperate mountain regions, the daily photoperiod and thermoperiod vary with the season, which can influence variation in the physiology and behavior of herbivores [10–12]. High temperatures can cause herbivores to reduce activity during the daytime and increase crepuscular activity [1, 13]. In general, herbivores can choose their resting and sheltering locations to experience more favorable temperature regimes, to avoid overheating [14], and to reduce heat loss [15, 16]. Thus, daily activity patterns of herbivores can demonstrate a clear seasonal variation, often with the lowest activity levels in winter [2, 17].

The takin (*Budorcas taxicolor*) is a large ungulate that inhabits temperate mountain regions and is considered to be a vulnerable species by the International Union for Conservation of Nature (IUCN) [18]. The animal is sexually dimorphic, and adult males are about 40% heavier

(Grant No. 31172112 and No. 31872252); the
Science and Technology Department of Shaanxi
Province (Grant No. 2015SZS-15-09 and
2018SZS-27, to Wenbo Yan); and the Special
Project for Biodiversity Protection of the Ministry of
Ecology and Environment, China No.
2017HB2096001006, to Zhigao Zeng. The funders
had no role in study design, data collection and
analysis, decision to publish, or preparation of the
manuscript

**Competing interests:** The authors have declared
that no competing interests exist.

than adult females [19]. This usually results in sexual differences in habitat and space use
among takins [20, 21], which is a pattern seen in many other ungulate species [22]. Some
researchers have hypothesized that sexual segregation among ungulates is the result of sexual
differences in activity patterns [23]. Captive Sichuan takins (*B. t. tibetana*) show seasonal and
sexual differences in activity budgets [24].

Because takins inhabit steep, complex forested alpine and subalpine areas, it is difficult to
explore their activity patterns and time budgets in detail, therefore there is little published infor-
mation on the activity patterns of takins [24–28]. Some studies suggest that the activity peak of
takins is during daytime in spring and summer; however, others show that takins have bimodal
crepuscular activity peaks [25]. Based on camera traps, takins showed three daily activity peri-
ods including midnight, early morning and mid-late afternoon, followed by three inactive peri-
ods in winter and spring [27]. However, previous research on takin activity patterns has had
shortcomings concerning the observation period and study method. The principal advantage of
global positioning system (GPS) radio collars over more traditional methods is the consistent
accrual of large numbers of locations per animal through automated tracking. Thus, GPS radio
collars have become an important wildlife research technology in recent years [29].

We studied the locomotor activity of takins in a temperate mountain region using GPS
radio collar technology. We predicted that (1) takins would show crepuscular locomotor activ-
ity peaks, and (2) seasonal and sexual differences in locomotor activity would be found in
accordance with seasonal variations in the photoperiod and thermoperiod. Additionally, we
predicted that microenvironment temperature and available food resources would be corre-
lated to locomotor activity because of seasonal altitudinal movements [30, 31].

## Materials and methods

### Study area

The study area is in and around the Foping National Nature Reserve (33°30′–33°50′ N, 107°
39′–107°58′ E), located in the Qinling Mountains, Shaanxi Province, China. Elevation ranges
between 810 m and 2904 m, and the area is characterized by rugged mountains (Fig 1). Based
on the temperature data of the Foping weather station around the study area from 1981 to 2016,
the lowest monthly mean temperature is -2.7°C in January; the highest monthly mean tempera-
ture is 28.3°C in July. Based on Zeng et al. [31], June–August is considered summer, Decem-
ber–March winter, with April–May and September–November forming the seasons of spring
and autumn, respectively. The cover types are mainly comprised of subalpine meadow, conifer
forests, mixed conifer-broadleaf forests, deciduous broadleaf forests and shrub [30, 32]. Except
golden takin (*B. t. bedfordi*), there are many other endangered mammals within the study area,
for instance, giant panda (*Ailuropoda melanoleuca*), Chinese goral (*Naemorhedus goral*), serow
(*Capricornis sumatraensis*), forest musk deer (*Moschus berezovskii*), Asiatic black bear (*Ursus
thibetanus*), golden snub-nosed monkey (*Rhinopithecus roxellana*) and golden cat (*Profelis tem-
mincki*). The Chinese goral's and serow's body size is much smaller than takin's, and they don't
compete with takins for food [33]. There are hardly any predators for takins since wild South
China tiger (*Pantheral tigris*) has been extinct in the study area [34].

### Takin data

We monitored ten adult takins (4 males and 6 females) tagged with GPS 7000M collars (Lotek
Wireless Inc., Ontario, Canada). Two animals were caught in 2013 and eight in 2014 (S1
Table). The dart rifle using immobilizing anesthetic was used to capture takins at a distance of
between 10 m and 20 m while the animals were congregated around a feeding site. The Xyla-
zine Hydrochloride Injection (Jilin Huamu Animal Health Product Co., Ltd., Changchun,

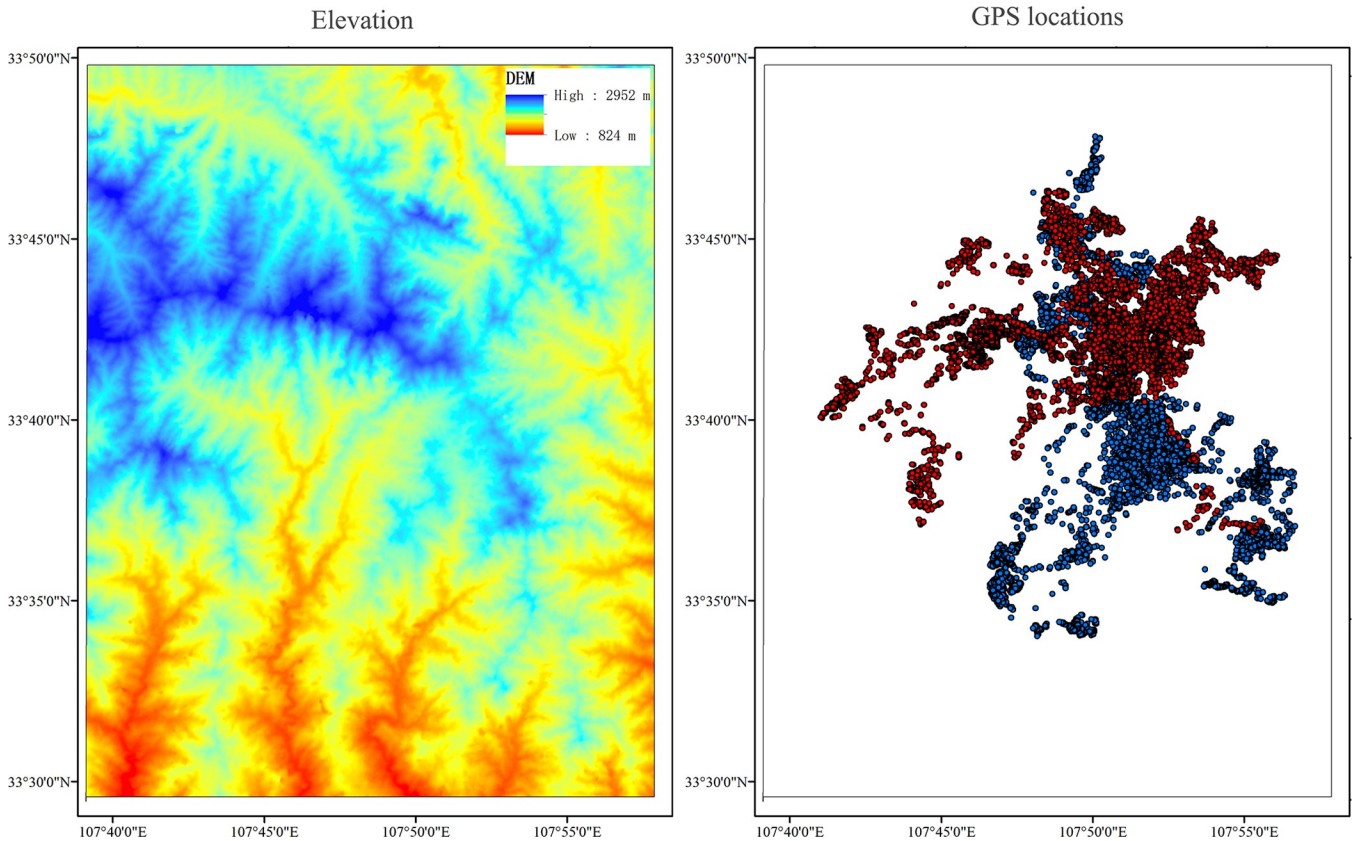

**Fig 1. The GPS locations of the 10 golden takins from 1 July 2014 to 30 June 2015, and elevation of the study area (Red points indicated locations of 6 female takins, blue points indicated locations of 4 male takins, digital elevation map data was downloaded from https://earthexplorer.usgs.gov/).**

China) was used as immobilizing anesthetic at a dose of 1.3–1.5 ml per 100 kg body mass. The Suxing Injection (Jilin Huamu Animal Health Product Co., Ltd., Changchun, China) was used as antidote to reverse the sedation at an equal dose of the anesthetic. An animal capture protocol of the study was approved by the Animal Ethics Committee of the Institute of Zoology, Chinese Academy of Sciences, and the National Forestry Agency of China (Linhuxuzhun # (2012)1630). The takin capture and collecting GPS data in the study area was permitted by the Foping national nature reserve administration.

The GPS 7000M collar weighted about 950 g, which was less than 1% of the body mass of takin. The collars were scheduled to acquire the position every two hours. The data information of each GPS position included latitude, longitude, temperature, dilution of precision (DOP), date and time. We periodically used a handheld command unit (HCU) to download data from the collars. Three-dimensional locations with a DOP < 10 were regarded as validated GPS locations while other less accurate locations were removed [35]. In these 10 collars, only 4 collars had still worked in June 2016; other collars did not work because of battery running out before June 2016. In order to keep statistical data unified, we used collar data from 1 July 2014 to 30 June 2015. During this period, we collected between 3075–4209 validated GPS locations for each animal (S1 Table). Using stationary collars, we estimated a location error of 10.73 m [21].

We separated takin movement paths into steps, which were defined as the true travel segments linking successive 2 h locations [36]. We first calculated the Euclidean distance between

successive locations (m). We subsequently calculated the true travel distance corrected using average change in altitude between successive locations [37]. Finally, we calculated movement rate by dividing the true travel distance by the time elapsed between them (h). Because movement rate can be influenced by time intervals [38], we used only GPS locations separated by 2 h (S1 Table). Based on Ensing et al. [37], movement rates from GPS successive locations were suitable as measures for daily locomotor activity in ungulates. Based on Zeng et al. [39], takins would mostly feed while walking. Therefore, movement rate would indicate feeding activity of takins.

## Period of day

In the study area, the sunrise shifts from 5:39 a.m. in June to 7:54 a.m. in January. Sunset shifts from 8:00 p.m. in June to 5:40 p.m. in December. We defined 'daytime' as the time from sunrise to sunset. 'Nighttime' was defined as the time between the end of nautical twilight in the evening and the start of nautical twilight in the morning. Because GPS locations are separated by a 2 h interval, locations at time t and t + 1 can be classified in different periods of the day. Thus, we classified steps into diurnal period if two successive GPS locations occurred during daytime, nocturnal period only if two successive GPS locations occurred during nighttime.

## Normalized difference vegetation index (NDVI)

As a browser species, takins forage on various species of plants, including mosses, ferns, herbs, shrubs and trees [40]. We used the Moderate Resolution Imaging Spectroradiometer (MODIS) NDVI (https://earthexplorer.usgs.gov/) as a surrogate for food abundance and availability. We calculated the NDVI value of takin GPS locations with the MODIS MOD13Q1 data (every 16 days at 250-m spatial resolution as a gridded level-3 product in the sinusoidal projection) using the extraction tools in ArcGIS 10.1 (Environmental System Research Institute Inc., Redlands, CA, USA). The mean NDVI of each movement step was calculated respectively as the means of its two endpoints NDVI.

## Microenvironment temperature

Because the takins inhabit steep, complex forested alpine and subalpine areas, we could not collect true ambient temperature of GPS locations. The temperature data taken by collars were mostly affected by the solar radiation, shade and other factors of GPS location microenvironment, although it also affected by the heat radiation of the takin wearing the collar. Therefore, the temperature of collar recorded could be as a surrogate for microenvironment temperature. The mean microenvironment temperature of each movement step was calculated respectively as the means of its two endpoints temperature from GPS collars.

## Statistical analysis

We used one-way ANOVA to compare seasonal differences on movement rates during daytime, nighttime and throughout the day. In order to see which determinants contribute to movement rate of the takins in each season, we constructed models including the factors (sex, temperature and NDVI) using the GLMM procedure, random factor being animal individual. All statistical analyses were performed using R version 3.5.1 [41].

# Results

There were apparent bimodal crepuscular movement peaks for the takins in each season (Fig 2). The movement peaks at dusk were especially obvious. The takin movement rates rapidly

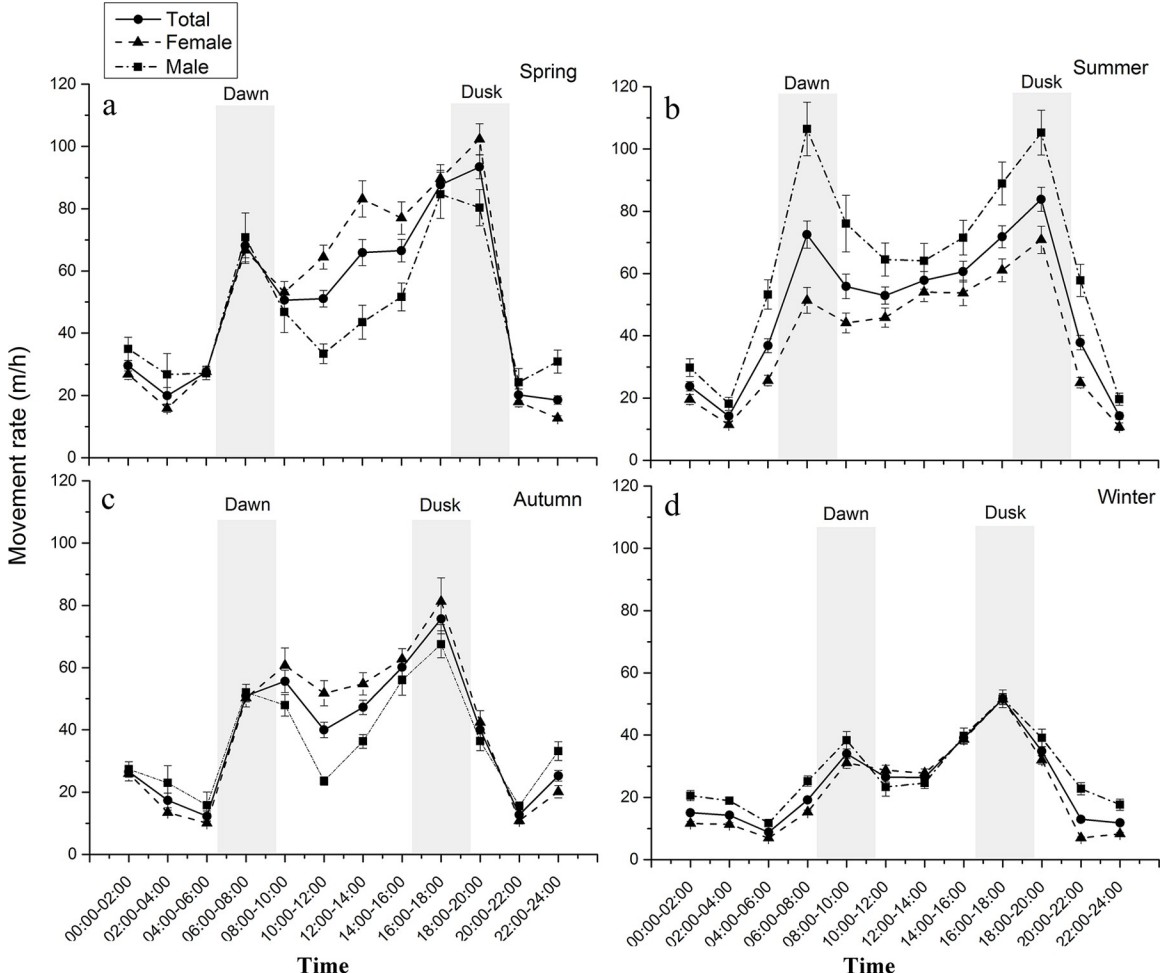

**Fig 2. Locomotor activity patterns of 10 takins monitored in the study area during the study period.** Dawn and dusk are shaded.

declined after dusk, and they had the slowest movement rates at night in each season (Fig 2). At dawn, their movement rates rapidly increased (Fig 2). After dawn, their movement rates also obviously declined. These resulted in an obvious movement peak at dawn (Fig 2).

The movement rate of the takins throughout the day and during daytime showed significant seasonal differences ($F_{3, 36} = 13.50$, $P < 0.01$ for whole day; $F_{3, 36} = 9.14$, $P < 0.01$ for daytime). The lowest movement rate of the takins throughout the day and during daytime was in winter (Fig 3). The differences in movement rate between winter and other there season were greater than location error of GPS collars. Although, the movement rate of the takins during nighttime showed seasonal difference ($F_{3, 36} = 3.15$, $P < 0.05$), the differences on movement rate between in winter and other there season were much less than location error of GPS collars (Fig 3).

The GLMM models showed that fixed factors (sex, temperature and NDVI) would affect movement rate of the takins (Table 1). The models also showed that correlations of these fixed factors were less than 0.65 in each season. In spring, the sexual effect estimate β via diurnal model showed significant negative effect on movement rate of the takins (Table 1). It indicated that female diurnal movement rate was significantly higher than that of males in spring. However, the sexual effect estimate β via nocturnal model showed significant positive effect on

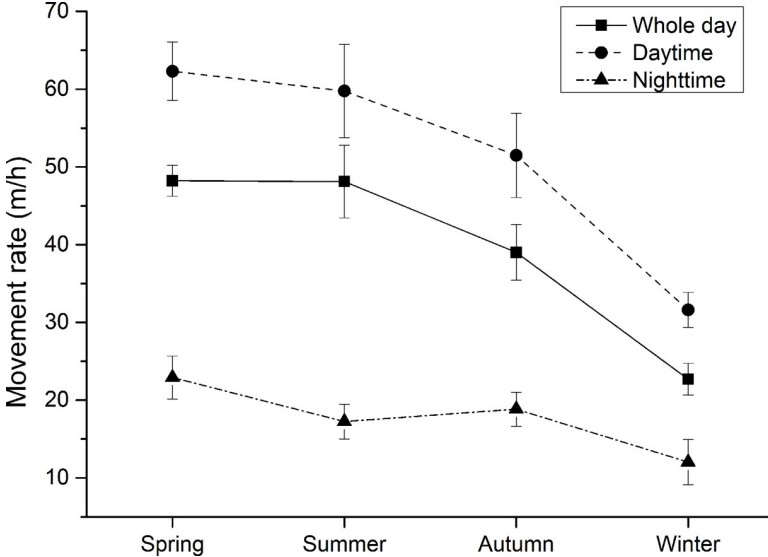

**Fig 3. Seasonal differences on movement rates of 10 takins during the study period.**

movement rate of the takins in spring (Table 1). It indicated that male nocturnal movement rate was significantly higher than that of females in spring. In summer, the sexual effect estimate β via whole day's model showed significant positive effect on movement rate of the takins (Table 1). It indicated that male movement rate was significantly higher than that of females in summer. In autumn, the sexual effect estimate β via nocturnal model showed significant positive effect on movement rate of the takins (Table 1). It indicated that male nocturnal movement rate was significantly higher than that of females in autumn. In winter, the sexual effect estimate β via model showed no effect on movement rate of the takins.

Microenvironment temperature of takin locations changed monthly, with higher temperatures during June to August and lower temperatures during December to February (Fig 4). The seasonal differences in takin movement rates were closely correlated to their changes. In summer, the takin movement rate was negatively correlated with microenvironment

**Table 1. Effect estimates β (SE) of predictor variables via model on takin movement rates in each season in the study area.**

| Season | Predictor/Fixed effect | Whole day β (SE) | Diurnal β (SE) | Nocturnal β (SE) |
|---|---|---|---|---|
| Spring | Sex/Male | -5.15(3.45) | **-15.92(5.35)** | **11.71(2.28)** |
| | Temperature | **1.05(0.21)** | **0.82(0.26)** | **-0.92(0.31)** |
| | NDVI | **34.93(7.87)** | **35.16(10.33)** | **35.85(9.46)** |
| Summer | Sex/Male | **17.65(6.92)** | **21.83(8.56)** | 6.36(3.83) |
| | Temperature | **-1.12(0.28)** | **-2.29(0.36)** | **-1.39(0.27)** |
| | NDVI | **38.99(6.13)** | **49.48(8.56)** | 6.31(5.32) |
| Autumn | Sex/Male | -4.27(6.99) | -12.24(9.83) | **7.67(3.27)** |
| | Temperature | 0.008(0.19) | -0.26(0.26) | -0.05(0.21) |
| | NDVI | **18.47(5.29)** | **19.81(7.57)** | 9.34(5.40) |
| Winter | Sex/Male | 5.80(4.01) | 4.31(4.20) | 7.18(5.17) |
| | Temperature | **-5.87(0.07)** | **-0.97(0.09)** | **-0.45(0.07)** |
| | NDVI | **-7.88(3.96)** | -11.85(6.25) | -5.85(3.31) |

Bold indicates significant effects.

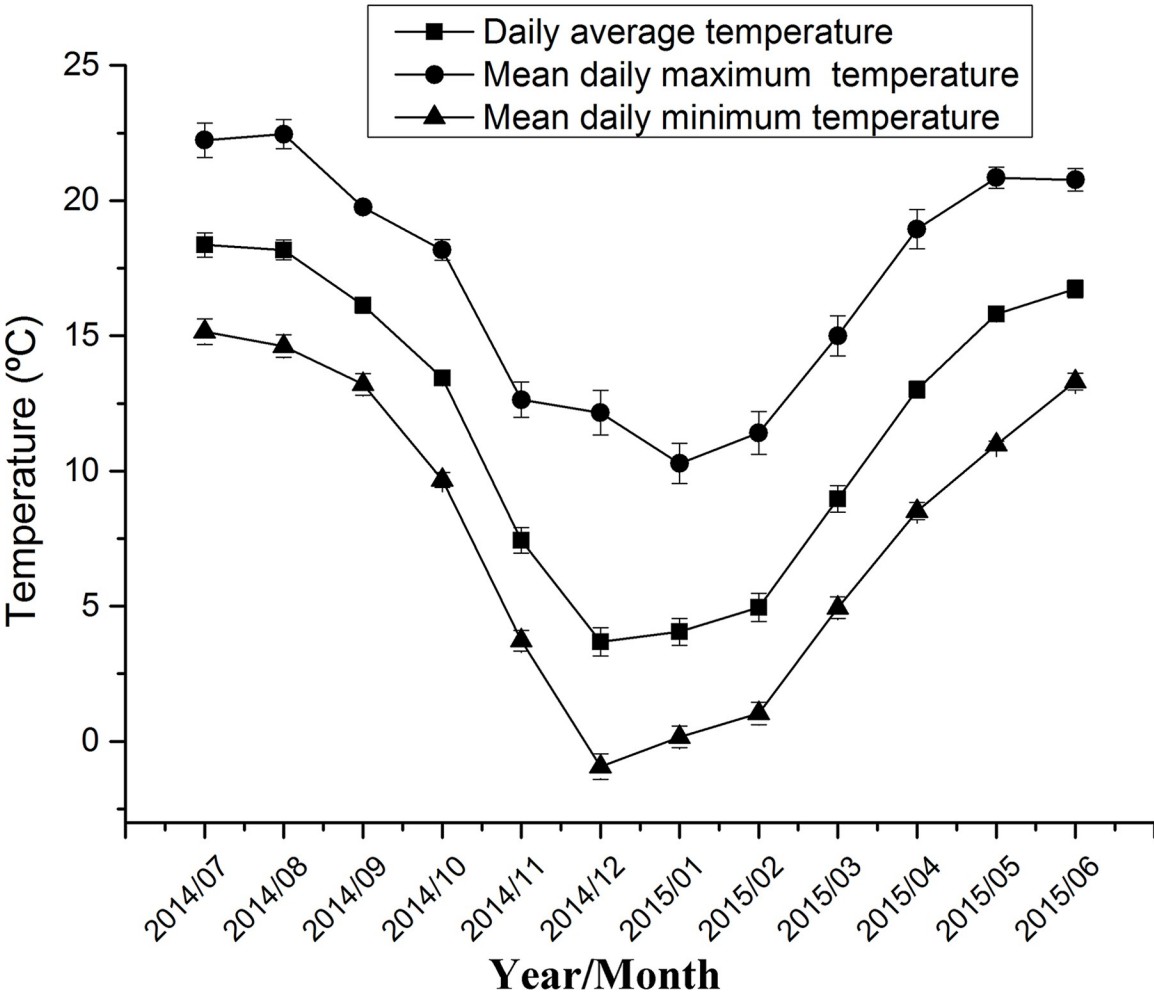

**Fig 4. Monthly changes in microenvironment temperature recorded by GPS collars during the study period.**

temperature, while there were highest temperatures. However, in winter, the takin movement rate was still negatively correlated with microenvironment temperature, while there were lowest temperatures. In spring, the takin movement rate was positively correlated with microenvironment temperature during daytime, but negatively correlated during nighttime. In autumn, the takin movement rate was not correlated with microenvironment temperature (Table 1). The takin movement rate was negatively correlated with NDVI in winter, but positively correlated in the other three seasons (Table 1).

## Discussion

This is the first quantitative analysis of locomotor activity patterns of *B. taxicolor*, an endangered species, based on data from GPS collars. This research showed that the takins had a bimodal crepuscular locomotor activity pattern, with an especially obvious movement peak at dusk. The takins showed significant seasonal and sexual differences in their movement rates. Our study provides evidence that microenvironment temperature and NDVI would affect movement rate of takins in each season.

Our results support the prediction that the takins have crepuscular locomotor activity peaks. Locomotor activity patterns of takins, which inhabit temperate mountain regions, were significantly influenced by the daily and seasonal variations in photoperiod and thermoperiod. Previous research has shown mostly crepuscular activity patterns in spring, summer and winter [25, 27]. However, our study showed that the takin movement peak was more obvious at dusk than at dawn (Fig 2). Many other wild ungulates show such crepuscular bimodal activity [42, 43]. The frequent alternations between periods of movement and rest during the 24 h were indicative of feeding and ruminating bouts, which are dependent on the morphological and physiological constraint set by the digestive system [44, 45].

The takins exhibited significant seasonal differences in their movement rates. The takins had the lowest movement rate in winter among the four seasons (Fig 3). The activity levels and movements of ungulates usually peak in summer and then decrease in winter [46, 47]. This might indicate a need to lower energy expenditure in winter, being the coldest period of the year, for individuals to conserve energy [48]. Ungulates have an alternative strategy for surviving harsh overwintering conditions [16], which is extensive basking at sunrise to rewarm after the nocturnal decrease in body temperature, without having to increase metabolic heat production. Consequently, the peak of locomotor activity of takins at dawn occurred much later in winter than in summer, and the winter dawn movement rate was significantly lower than in summer (Fig 2). Moreover, the takin movement rate was negatively correlated with microenvironment temperature during daytime in winter (Table 1), indicating that the takins decrease their movements with increasing microenvironment temperature. The takins prefer to lie down and bask in the sun during in this period and usually select south-facing slopes with higher solar radiation [31, 49]. During nighttime in winter, the takin movement rate was also negatively correlated with microenvironment temperature (Table 1), indicating that takins search for warm night habitats to avoid being exposed to the cold. Maintaining thermal balance is most important for the takins to select suitable habitats in winter [31]. In winter, when the availability of food for the takins is lowest, the negative correlation between the takin movement rate and NDVI (Table 1) also indicates that searching for forage is less important than other requirements, such as conserving energy.

The foraging movement of alpine ungulates is usually influenced by plant biomass in different seasons [50]. In spring, the microenvironment temperature of the takin habitat increased quickly (Fig 4) and their effect estimate β on movement rates was also maximum (Table 1). Spring is the migrating season for takins to descend into low-altitude regions [30]. Generally, takins move to regions with warm, new green vegetation and higher quality forage in this period [31]. By following spatiotemporal patterns in new plant growth via migration between seasonal ranges, migratory ungulates are predicted to enhance rates of energy intake [51]. Therefore, the positive correlation between the movement rate and NDVI indicated that the takins could forage more nutritious and abundant food during daytime in spring (Table 1). In summer, temperatures are at their highest and takins move upward to forage in high altitude regions [31]. Though temperatures in summer were highest during the year (Fig 4), the takin movement rate was negatively correlated with microenvironment temperature during daytime (Table 1), indicating that the animals prefer to inhabit a cooler environment. The decrease in activity level in response to microenvironment temperature as a strategy against over-heating has previously been researched in ungulates [52]. However, we know that takins also look for abundant forage to maximize their energy acquisition in summer, based on the positive correlation between the movement rate and NDVI (Table 1). This explanation is accordance with the forage abundance hypothesis [53]. Generally, ungulates would spend more time searching for and selecting higher quality forage when forage abundance is high [54]. In summary, microenvironment temperature and NDVI had strong effects on takin movements.

The takins also exhibited significant sexual differences in their movement rates. Summer is the mating season for the species [55]. We may conjecture that higher movement rate of males compared to females in summer is closely related to their mating behavior. Male takins commonly adopt visual displays in inter-sexual interaction during the mating season. Males can effectively alter their search efforts via faster movement rates during the mating season [56]. Therefore, male takins increased their locomotor activity traveling wider areas in order to increase their reproductive opportunities in summer [28, 55]. Spring is an important season for female takins to feed calves [49]. Females with offspring need to search widely for high quality food to supply their high energy expenditure during lactation, which is the most likely reason why the female movement rates during daytime were higher than those of males in spring. Because of calf limited vision and locomotor activity, females with offspring would be expected to be less mobile during nighttime. Our results also showed that nocturnal locomotor activity of females was lower than those of males in spring. Autumn is also an important season for female takins to forage and reserve energy before the bitter winter. Reserves of energy are very important for female pregnancy and birth in winter. Accordingly, takin movement rates were positive correlation with NDVI in autumn. Female takins would be pregnant and give birth in winter [49]. Females would select warm and shelter birth sites and care for her calves. Therefore, females would be less mobile than males in winter. Our results demonstrated that locomotor activity of females was lower than those of males in winter. To sum up, breeding behavior and the feeding of calves have important effects on the sexual differences in movement rates of takins.

Because our sampling rate (12 values per day) was lower, we could not accurately analyze behavior activity of the takins. However, our results provide new information about locomotor activity patterns of takins experiencing complex forested alpine and subalpine temperate environmental conditions. First, we confirmed the prediction that both photoperiod and thermoperiod shape the bimodal pattern, with peaks at dawn and dusk for large ungulates inhabiting temperate mountain regions. Second, we present evidence that movement rate of takins would be correlated to sex, microenvironment temperature and NDVI, and by implication available food resources in each season. Third, we confirmed the prediction that breeding behavior and calves feeding have important effects on sexual differences in the movement rates of takins.

## Supporting information

**S1 Table. Takin-specific GPS collar data including individual ID, years estimated age, sex, estimated body mass, start dates of collar deployment, number of fixes after data filtering and number of data separated only by 2 h from 1 July 2014 to 30 June 2015.**
(DOCX)

**S2 Table. Takin-specific GPS collar data including individual ID, location ID, date, time, latitude, longitude, height, DOP and temperature from 1 July 2014 to 30 June 2015.**
(XLSX)

## Acknowledgments

We appreciate Foping Nature Reserve for the management support. We especially thank the reserve staff for their field work. We also thank LetPub (www.letpub.com) for its linguistic assistance during the preparation of this manuscript.

## Author Contributions

**Conceptualization:** Wenbo Yan.

**Formal analysis:** Wenbo Yan.

**Funding acquisition:** Zhigao Zeng.

**Investigation:** Yan Duan.

**Methodology:** Huisheng Gong.

**Project administration:** Leigang Zhao.

**Resources:** Aliu Peng.

**Supervision:** Zhigao Zeng.

**Writing – original draft:** Wenbo Yan.

**Writing – review & editing:** Zhigao Zeng.

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
