## [Decision Letter · Decision Letter 0]

21 Oct 2019

PONE-D-19-21970

Locomotor activity patterns of takin (Budorcas taxicolor) in a temperate mountain region

PLOS ONE

Dear Dr. Wenbo Yan,

Thank you for submitting your manuscript to PLOS ONE. After careful consideration, we feel that it has merit but does not fully meet PLOS ONE’s publication criteria as it currently stands. Therefore, we invite you to submit a revised version of the manuscript that addresses the points raised during the review process.

In particular, please make sure you comprehensively address the concerns of reviewer 1 regarding methodological choices, possible sources of error and data accessibility, of reviewer 2 regarding the statistical analysis, and of both reviewers regarding the clarity of language.

We would appreciate receiving your revised manuscript by December 2, 2019. To enhance the reproducibility of your results, we recommend that if applicable you deposit your laboratory protocols in protocols.io, where a protocol can be assigned its own identifier (DOI) such that it can be cited independently in the future. For instructions see: http://journals.plos.org/plosone/s/submission-guidelines#loc-laboratory-protocols

We look forward to receiving your revised manuscript.

Kind regards,

Julia Molnar, PhD

Academic Editor

PLOS ONE

**Journal Requirements:**

2. We note that  Figure(s) 1 in your submission contain [map/satellite] images which may be copyrighted. All PLOS content is published under the Creative Commons Attribution License (CC BY 4.0), which means that the manuscript, images, and Supporting Information files will be freely available online, and any third party is permitted to access, download, copy, distribute, and use these materials in any way, even commercially, with proper attribution. For these reasons, we cannot publish previously copyrighted maps or satellite images created using proprietary data, such as Google software (Google Maps, Street View, and Earth). For more information, see our copyright guidelines: http://journals.plos.org/plosone/s/licenses-and-copyright.

a) You may seek permission from the original copyright holder of Figure(s) [#] to publish the content specifically under the CC BY 4.0 license.  

**Comments to the Author**

1. Is the manuscript technically sound, and do the data support the conclusions?

Reviewer #1: Partly

Reviewer #2: Partly

2. Has the statistical analysis been performed appropriately and rigorously? 

Reviewer #1: I Don't Know

Reviewer #2: No

3. Have the authors made all data underlying the findings in their manuscript fully available?

Reviewer #1: No

Reviewer #2: Yes

4. Is the manuscript presented in an intelligible fashion and written in standard English?

Reviewer #1: No

Reviewer #2: No

5. Review Comments to the Author

Reviewer #1: The manuscript describes the locomotion activity pattern of 10 free-living takins that were tagged with GPS collars. The data were correlated with NDVI data and temperature data measured by the GPS-collar.

In my opinion, the study gives no new insight into activity behavior of herbivores in temperated area; former studies which were carried out by camera traps have shown similar results and the hypotheses here (which are not daring anyway) were -as expected- simply confirmed. In addition, the technique used was not exploited in all its possibilities:

1) The authors used GPS-collars from Lotek company. Lotek collars also have 3D-activity loggers; These are much better suited to continuously and high-resolution record the activity of the animals and sometimes even to distinguish between different behaviors (Brown et al. 2014 in Animal Biotelemetry).

2) The spatial data (positions) measured by the GPS collars were not used in spatial analyses (home range calculation, habitat use).

3) Nowadays, linking activity data (measured by accelerometers) to spatial data (GPS positions) gives new insight into animal behavior (what is an animal doing and at what place?) but in this study, spatial data were used to roughly extrapolate locomotor activity data.

The study has some other significant shortcoming:

1) The GPS recording interval of 2 h (as a minimum) is still very rough for locomotion activity data. The cited study (Ensing et al. 2014) which proves that GPS positions can be used to deduce the activity behavior has been used data measured every hour (wich is twice as much data per day like here). Furthermore, Ensing et al. 2014 did include (spatial) error corrections in their calculations; in the manuscript here I did not find any of this.

2) In the manuscript, temperature data taken by the collars were defined as the ambient temperature but these temperatures are significantly affected by the heat radiation of the animal wearing the collar, the solar radiation and others influenced. Initially, these temperature sensors were used to determine whether the animal wearing the collar died, causing a drop in the measured temperature and much less daily fluctuations. Thus, a clear measurment of the ambient temperature by the collars is not proven so far. Figure 6 of the manuscript confirms my statement: Monthly mean temperatures measured by the collar (Fig. 6) are quite different from the monthly mean temperatures for January and July given in line 69/70. Before using the temperature data from the collars weared by an animal as ambient temperature data, one should have evaluate their real relationship to the ambient temperatures .

3) In the manuscript, bimodal activity patterns and different seasonal activity levels seem to be caused only by the ambient temperature and the food supply (NDVI) as only these parameters are discussed. Other basic physiological (clock-gene, circadian evolutionary adaptations) and ecological (predators, food competitors) influencing factors are not even mentioned throughout the manuscript.

4) Most analyzes (for instance reference number 34 or the control charts) are quite old-school and - in contrast - badly presented that sometimes I could not fully understand them.

5) The measured data are not fully available and - in oposite to the author's statement - also not available in the manuscript. This makes the study incomprehensible for other researchers. For instance, GPS data could be stored in a data base (like Movebank which has a lot of data measured by Lotek collars) from which they are available in some cases also shareable .

6) English language should be improved, sometimes I couldn´t understand and it was difficult to read.

Additionally some comments to special lines in the manuscript:

line 31: insert a space after "loss"

line 47: studies do not think; maybe replace "think" with "show".

line 70: why general weather data are quoted here from other papers about takins? There are general weather stations whose measurements should serve as a source.

line 73-79: I miss information about nutrition competitors or predators for the takins.

Figure 1: I miss the spatial position of the study area within China or at least longitudes and latitudes. It would be better to show the different individuals (or at least genders) with different colors.

line 88: Were the animals lured into traps beforehand or how was it possible to get to the animals in a shooting distance?

line 90/91 and 96/97: What is body mass of the takins? Please insert their individual body mass in S1 table.

line 91 and line 97: write "body" before "mass"

line 99: "et al." irritates me here.

line 100: change "in" to "from" (download data FROM the collar)

line 114-121: Which twilight was taken? Once they write about sunrise / sunset (sun stands 0 degrees below the horizon), once about nautical twiligth (6-12 degrees below horizon). Why is a "crepuscular period" explained which does not appear later in the manuscript?

line 150: add an "o" in the first word.

line 152: "to the form" - what does that mean?

Fig. 2-5: I would prefer to present the mean daily activity curve (for 24 hours with standard deviation) per season and per sex in which the twilight times are either hatched or marked by arrows. So it would be only half as many diagrams (8 pieces) but more information in it (the whole activity patterns over the entire day).

line 168-174 & table 1: I do not understand that: In the text (line 168, 170, 173) is spoken by "moving rates", in the table the "effect estimates" are given. Where I can you see the exact formulas, results and significance results of the calculated GLMs? I do not understand this table.

line 207: delete "for foraging" because you do not know if it was really foraging.

223: Reference 48 is the same as Reference 17

Discussion: Overall, it seems to me that too much is being interpreted into the data. Finally, distances have been measured which the collared animal travelled within 2 hours. There is no information on the behavior of the animal at the measuring points or during the time in between and there are a maximum of 12 measuring points per day. Here, these data are often discussed as if one had measured the foraging behavior but there is no proof for it.

287/288 & 17/18: I'm not quite sure what kind of "efforts" were informed by this study. Please, could you precise this statement!

375: note: "written in Chinese"

412: delete "a" in activity

417: Delete "c" in charts

441-443: Delete it as it is the same reference like numer 17

S1 table: Where are the end dates of collar deployment (like written in the table description)? Add to "r" to "Start" in the ehead of column 4! Would be happy to add the body mass of the individual during collaring. Add "[years]" in the head of column 2. You also have the method of age estimation!

Reviewer #2: Review

PONE-D-19-21970

Locomotor activity patterns of takin (Budorcas taxicolor) in a temperate mountain

Region

This study is about the locomotor activity patterns of takin and the factors shaping those patterns. The authors show a clear sexual dimorphism in movement rates during the seasons and further provide evidence for a connection between takin movement rates and resource availability. Since this study is the first to employ GPS tracking to observe takin movement it provides new and detailed insights into the behavioural adaptations of this species to ecological conditions, intrinsic requirements and life history. Especially since the study species is highly endangered the conservational value of this study should be emphasized.

However, I have some concerns with the manuscript in its current form that need to be addressed:

1) The statistical analysis raises the concern of over fitted generalized linear models which weakens the results and makes me wonder if you can trust them at all. Calculating GLMs with four predictor variables on a sample size of ten is unwise, because the over fitting of the model increases the chances that your regression coefficients represent the noise rather than the genuine relationships in the population. Additionally, over fitting reduces the generalizability outside the original dataset, which is, especially in the context of the conservational value of the study, problematic. Besides the over fitting of the model, the chosen method itself might not be entirely correct since based on the nature of data sampling there is potential for some variation between the individuals that might need to be addressed. Imploring generalised linear mixed models that incorporate for example the differing durations of collar deployment of the animals in their random structure might be a better choice.

I would advise the authors to thoroughly rethink their statistical analysis and revise the manuscript accordingly.

2) The whole manuscript needs to be revised language wise. Especially parts of the introduction and discussion are hard to follow because language errors distract from the actual text. This is not a major issue and can, in my opinion, be easily fixed. I did comment on a few of the language problems (see below), but would advise the authors to have the manuscript proof read by a native speaker if possible.

I have some minor concerns which are addressed in the following:

Line 9: Movement rates can not show differences, the animals or sexes can show differences in their movement rates.

Line 11-13: This sentence does not make sense and is grammatically incorrect. Please rewrite it.

Line 21-23: I think you mean ”Investigating locomotor activity patterns”. The patterns themselves can not reveal anything, only if they are looked at the and put together with other factors can they reveal information.

Line 55: change “recently” to “recent”

Line 68: change „2,904“ to „2.904“

Line 87-88: Is the unbalance of individuals being collard in different years and the resulting differences in the duration of collaring problematic? Please explain how you approached that.

Line 96-97: Were the collars removed after the tracking period? How many collars were retrieved?

Line 98: delete „was“

Line 99: delete the comma after date and the “et al.”

Line 109: In your supplement table S1 I can not see the end dates of collar deployment for each individual, although your table header says that they should be in there. Does that mean that animals were collared for close to, or in some cases over a whole year?

Lines 137-147: I am concerned with your statistics, the way I understand it you calculated models for each response variable with the same four fixed factors (sex, season, temperature and NDVI), based on your rather small sample size of 10 animals I think your models are over fitted. Based on the common rule of thumb for linear regressions, there should be 10 observations per predictor, which would reduce the number of possible predictors in your models drastically.

Furthermore, since your animals were collared in two different years and therefore had different durations of wearing the collars it might be wise to incorporate that potential source of variation into your models by calculating GLMMs instead of GLMs.

Line 151: returned to what form?

Line 171-172: “throughout day and during daytime” is the same thing, you do not need this doubling, or does during day mean during the whole period of 24 hours?

Line 172: contrary to the movement rates at night-time which were higher for males

Table 1: Predictor and fixed effects are the same thing, what you are showing in the column “fixed effects” are the factor levels of each predictor/fixed effect.

Line 183: delete “both”

Line 234: “for cold” does not make sense, I think you mean that they retreat to warmer resting places to avoid being exposed to the cold.

Line 279: change “pregnancy” to “pregnant”

Line 280: infant is a rather anthrophonic terminus, maybe use juvenile, calve or young.

6. PLOS authors have the option to publish the peer review history of their article (what does this mean?). If published, this will include your full peer review and any attached files.

Reviewer #1: No

Reviewer #2: No

---

## [Author Response · Author response to Decision Letter 0]

27 Mar 2020

Reviewer #1: The manuscript describes the locomotion activity pattern of 10 free-living takins that were tagged with GPS collars. The data were correlated with NDVI data and temperature data measured by the GPS-collar.

In my opinion, the study gives no new insight into activity behavior of herbivores in temperated area; former studies which were carried out by camera traps have shown similar results and the hypotheses here (which are not daring anyway) were -as expected- simply confirmed. In addition, the technique used was not exploited in all its possibilities:

1) The authors used GPS-collars from Lotek company. Lotek collars also have 3D-activity loggers; These are much better suited to continuously and high-resolution record the activity of the animals and sometimes even to distinguish between different behaviors (Brown et al. 2014 in Animal Biotelemetry).

I’m sorry. Because of project capital limitation, the Lotak collars GPS7000 of we used were not fitted out with 3D-activity loggers.

2) The spatial data (positions) measured by the GPS collars were not used in spatial analyses (home range calculation, habitat use).

OK. We had used the spatial data (positions) measured by the GPS collars to analyses home range and habitat use of takin. 

Yan WB, Zeng ZG, Gong HS, He XB, Liu XY, Si KC, et al. Habitat use and selection by takin in the Qinling Mountains, China. Wildlife Research. 2016; 43(8):671-80.

Yan WB, Zeng ZG, Gong HS, He XB, Liu XY, Ma YS, et al. Seasonal variation and sexual difference of home ranges by takins. The Journal of Wildlife Management. 2017; 81(5):938-42.

3) Nowadays, linking activity data (measured by accelerometers) to spatial data (GPS positions) gives new insight into animal behavior (what is an animal doing and at what place?) but in this study, spatial data were used to roughly extrapolate locomotor activity data.

OK. Because of project capital limitation, the Lotak collars GPS7000 of we used were not fitted out with 3D-activity loggers. The previous research on takin activity has had shortcomings concerning the observation period and study method. Fox example, former studie which were carried out by (Li MF et al. 2011 in Sichuan Journal of Zoology) was researched using only 6 camera traps in winter and spring. Therefore, our study first researched locomotion activity pattern of takin thourghout year, seasonal and sexual differrences. 

The study has some other significant shortcoming:

1) The GPS recording interval of 2 h (as a minimum) is still very rough for locomotion activity data. The cited study (Ensing et al. 2014) which proves that GPS positions can be used to deduce the activity behavior has been used data measured every hour (which is twice as much data per day like here). Furthermore, Ensing et al. 2014 did include (spatial) error corrections in their calculations; in the manuscript here I did not find any of this.

OK.We accept this. We calculated the true travel distance corrected using average change in altitude between successive locations based on Ensing et al. (2014). 

2) In the manuscript, temperature data taken by the collars were defined as the ambient temperature but these temperatures are significantly affected by the heat radiation of the animal wearing the collar, the solar radiation and others influenced. Initially, these temperature sensors were used to determine whether the animal wearing the collar died, causing a drop in the measured temperature and much less daily fluctuations. Thus, a clear measurment of the ambient temperature by the collars is not proven so far. Figure 6 of the manuscript confirms my statement: Monthly mean temperatures measured by the collar (Fig. 6) are quite different from the monthly mean temperatures for January and July given in line 69/70. Before using the temperature data from the collars weared by an animal as ambient temperature data, one should have evaluate their real relationship to the ambient temperatures .

OK.We accept this. We have evaluate their real relationship to the ambient temperatures.

3) In the manuscript, bimodal activity patterns and different seasonal activity levels seem to be caused only by the ambient temperature and the food supply (NDVI) as only these parameters are discussed. Other basic physiological (clock-gene, circadian evolutionary adaptations) and ecological (predators, food competitors) influencing factors are not even mentioned throughout the manuscript.

OK. We accept this. We explained the effect of predators and food competitors for takin. 

4) Most analyzes (for instance reference number 34 or the control charts) are quite old-school and - in contrast - badly presented that sometimes I could not fully understand them.

OK. We accept this. We had present the mean daily activity curve (for 24 hours with standard deviation) per season and per sex in which the twilight times was shaded.

5) The measured data are not fully available and - in oposite to the author's statement - also not available in the manuscript. This makes the study incomprehensible for other researchers. For instance, GPS data could be stored in a data base (like Movebank which has a lot of data measured by Lotek collars) from which they are available in some cases also shareable .

OK. We accept this. We had present Takin-specific GPS collar data including individual ID, location ID, date, time, latitude, longitude, height, DOP and temperature from 1 July 2014 to 30 June 2015 in S2_table.

6) English language should be improved, sometimes I couldn´t understand and it was difficult to read.

OK. We accept this.

Additionally some comments to special lines in the manuscript:

line 31: insert a space after "loss"

OK. We accept this.

line 47: studies do not think; maybe replace "think" with "show".

OK. We accept this.

line 70: why general weather data are quoted here from other papers about takins? There are general weather stations whose measurements should serve as a source.

OK. We accept this. We used the temperature data of Foping weather station around the study area from 1981 to 2010.

line 73-79: I miss information about nutrition competitors or predators for the takins.

OK. We accept this. We increased information about nutrition competitors or predators for the takins.

Figure 1: I miss the spatial position of the study area within China or at least longitudes and latitudes. It would be better to show the different individuals (or at least genders) with different colors.

OK. We accept this.

line 88: Were the animals lured into traps beforehand or how was it possible to get to the animals in a shooting distance?

OK. We accept this. The dart rifle using immobilizing anesthetic was used to capture takins at a distance of between 10 and 20 meters while the animals were congregated around a feeding site.

line 90/91 and 96/97: What is body mass of the takins? Please insert their individual body mass in S1 table.

OK. We accept this.

line 91 and line 97: write "body" before "mass"

OK. We accept this.

line 99: "et al." irritates me here.

OK. We deleted “et al.”

line 100: change "in" to "from" (download data FROM the collar)

OK. We accept this.

line 114-121: Which twilight was taken? Once they write about sunrise / sunset (sun stands 0 degrees below the horizon), once about nautical twiligth (6-12 degrees below horizon). Why is a "crepuscular period" explained which does not appear later in the manuscript?

OK. We redefined four period of day: dawn, daytime, dusk and nighttime.

line 150: add an "o" in the first word.

OK. We accept this.

line 152: "to the form" - what does that mean?

OK. We revised these results.

Fig. 2-5: I would prefer to present the mean daily activity curve (for 24 hours with standard deviation) per season and per sex in which the twilight times are either hatched or marked by arrows. So it would be only half as many diagrams (8 pieces) but more information in it (the whole activity patterns over the entire day).

OK. We accept this. We had present the mean daily activity curve (for 24 hours with standard deviation) per season and per sex in which the twilight times was shaded.

line 168-174 & table 1: I do not understand that: In the text (line 168, 170, 173) is spoken by "moving rates", in the table the "effect estimates" are given. Where I can you see the exact formulas, results and significance results of the calculated GLMs? I do not understand this table.

OK. We revised these results.

line 207: delete "for foraging" because you do not know if it was really foraging.

OK. We accept this.

223: Reference 48 is the same as Reference 17

OK. We accept this.

Discussion: Overall, it seems to me that too much is being interpreted into the data. Finally, distances have been measured which the collared animal travelled within 2 hours. There is no information on the behavior of the animal at the measuring points or during the time in between and there are a maximum of 12 measuring points per day. Here, these data are often discussed as if one had measured the foraging behavior but there is no proof for it.

Because takins inhabit steep, complex forested alpine and subalpine areas, it is difficult to explore their activity patterns and time budgets in detail, and therefore there is little published information on the activity patterns of takins. Based on our previous research, takins would mostly forage wihle walking (Zeng et al. 2001. Chinese Journal of Zoology.). 

287/288 & 17/18: I'm not quite sure what kind of "efforts" were informed by this study. Please, could you precise this statement!

OK. We deleted this statement.

375: note: "written in Chinese"

OK. We accept this.

412: delete "a" in activity

OK. We accept this.

417: Delete "c" in charts

OK. We accept this.

441-443: Delete it as it is the same reference like numer 17

OK. We accept this.

S1 table: Where are the end dates of collar deployment (like written in the table description)? Add to "r" to "Start" in the ehead of column 4! Would be happy to add the body mass of the individual during collaring. Add "[years]" in the head of column 2. You also have the method of age estimation!

OK. We accept this. Some GPS collars did not work one whole year after 30 June 2015. For keeping data consistent, we used collar data from 1 July 2014 to 30 June 2015. The end dates of collar data was 30 June 2015 (like written in the table description). Therefore, we did not list in the table. We could estimate age of takin based on their degrees of wear.

Reviewer #2: Review

Locomotor activity patterns of takin (Budorcas taxicolor) in a temperate mountain region

This study is about the locomotor activity patterns of takin and the factors shaping those patterns. The authors show a clear sexual dimorphism in movement rates during the seasons and further provide evidence for a connection between takin movement rates and resource availability. Since this study is the first to employ GPS tracking to observe takin movement it provides new and detailed insights into the behavioural adaptations of this species to ecological conditions, intrinsic requirements and life history. Especially since the study species is highly endangered the conservational value of this study should be emphasized.

However, I have some concerns with the manuscript in its current form that need to be addressed:

1) The statistical analysis raises the concern of over fitted generalized linear models which weakens the results and makes me wonder if you can trust them at all. Calculating GLMs with four predictor variables on a sample size of ten is unwise, because the over fitting of the model increases the chances that your regression coefficients represent the noise rather than the genuine relationships in the population. Additionally, over fitting reduces the generalizability outside the original dataset, which is, especially in the context of the conservational value of the study, problematic. Besides the over fitting of the model, the chosen method itself might not be entirely correct since based on the nature of data sampling there is potential for some variation between the individuals that might need to be addressed. Imploring generalised linear mixed models that incorporate for example the differing durations of collar deployment of the animals in their random structure might be a better choice.

I would advise the authors to thoroughly rethink their statistical analysis and revise the manuscript accordingly.

OK. We accept this. We reconstructed models including the factors (sex, season, temperature and NDVI) using the GLMM procedure, and revise the manuscript accordingly.

2) The whole manuscript needs to be revised language wise. Especially parts of the introduction and discussion are hard to follow because language errors distract from the actual text. This is not a major issue and can, in my opinion, be easily fixed. I did comment on a few of the language problems (see below), but would advise the authors to have the manuscript proof read by a native speaker if possible.

OK. We accept this.

I have some minor concerns which are addressed in the following:

Line 9: Movement rates can not show differences, the animals or sexes can show differences in their movement rates.

OK. We accept this.

Line 11-13: This sentence does not make sense and is grammatically incorrect. Please rewrite it.

OK. We accept this.

Line 21-23: I think you mean ”Investigating locomotor activity patterns”. The patterns themselves can not reveal anything, only if they are looked at the and put together with other factors can they reveal information.

OK. We accept this. We revised this paragraph.

Line 55: change “recently” to “recent”

OK. We accept this.

Line 68: change „2,904“ to „2.904“

OK. We accept this.

Line 87-88: Is the unbalance of individuals being collard in different years and the resulting differences in the duration of collaring problematic? Please explain how you approached that.

Although takins were collard in different years, we only used all collar data from 1 July 2014 to 30 June 2015, other collar data was not used in this research.

Line 96-97: Were the collars removed after the tracking period? How many collars were retrieved?

The collars still worked after 30 June 2015, but only six collars worked untill 30 June 2015, other four collars did not work one whohe year 30 June 2015. Therefore, we only used all collar data from 1 July 2014 to 30 June 2015. Finally, we retrieved eight collars, other two lose.

Line 98: delete „was“

OK. We accept this.

Line 99: delete the comma after date and the “et al.”

OK. We accept this.

Line 109: In your supplement table S1 I can not see the end dates of collar deployment for each individual, although your table header says that they should be in there. Does that mean that animals were collared for close to, or in some cases over a whole year?

The takins were collared over one whole year, but four collars did not work one whohe year 30 June 2015. Therefore, we only used all collar data from 1 July 2014 to 30 June 2015.

Lines 137-147: I am concerned with your statistics, the way I understand it you calculated models for each response variable with the same four fixed factors (sex, season, temperature and NDVI), based on your rather small sample size of 10 animals I think your models are over fitted. Based on the common rule of thumb for linear regressions, there should be 10 observations per predictor, which would reduce the number of possible predictors in your models drastically.

Furthermore, since your animals were collared in two different years and therefore had different durations of wearing the collars it might be wise to incorporate that potential source of variation into your models by calculating GLMMs instead of GLMs.

OK. We accept this. We reconstructed models including the factors (sex, season, temperature and NDVI) using the GLMM procedure, and revise the manuscript accordingly.

Line 151: returned to what form?

OK. We revised these results.

Line 171-172: “throughout day and during daytime” is the same thing, you do not need this doubling, or does during day mean during the whole period of 24 hours?

OK. We revised these results.

Line 172: contrary to the movement rates at night-time which were higher for males

OK. We revised these results.

Table 1: Predictor and fixed effects are the same thing, what you are showing in the column “fixed effects” are the factor levels of each predictor/fixed effect.

OK. We revised this table.

Line 183: delete “both”

OK. We accept this.

Line 234: “for cold” does not make sense, I think you mean that they retreat to warmer resting places to avoid being exposed to the cold.

OK. We accept this.

Line 279: change “pregnancy” to “pregnant”

OK. We accept this.

Line 280: infant is a rather anthrophonic terminus, maybe use juvenile, calve or young.

OK. We accept this.

---

## [Decision Letter · Decision Letter 1]

22 Apr 2020

PONE-D-19-21970R1

Locomotor activity patterns of takin (Budorcas taxicolor) in a temperate mountain region

PLOS ONE

Dear Dr. Wenbo Yan,

Thank you for submitting your manuscript to PLOS ONE. After careful consideration, we feel that it has merit but does not fully meet PLOS ONE’s publication criteria as it currently stands. Therefore, we invite you to submit a revised version of the manuscript that addresses the points raised during the review process.

Both reviewers note that the manuscript has been substantially improved. However, both still have serious concerns about the statistical tests. Please be sure to address these in your revision. In addition, please consider Reviewer 1's concerns about measurement error.

We would appreciate receiving your revised manuscript by May 29, 2020. To enhance the reproducibility of your results, we recommend that if applicable you deposit your laboratory protocols in protocols.io, where a protocol can be assigned its own identifier (DOI) such that it can be cited independently in the future. For instructions see: http://journals.plos.org/plosone/s/submission-guidelines#loc-laboratory-protocols

We look forward to receiving your revised manuscript.

Kind regards,

Julia Molnar, PhD

Academic Editor

PLOS ONE

Reviewers' comments:

Reviewer's Responses to Questions

**Comments to the Author**

1. If the authors have adequately addressed your comments raised in a previous round of review and you feel that this manuscript is now acceptable for publication, you may indicate that here to bypass the “Comments to the Author” section, enter your conflict of interest statement in the “Confidential to Editor” section, and submit your "Accept" recommendation.

Reviewer #1: (No Response)

Reviewer #2: (No Response)

2. Is the manuscript technically sound, and do the data support the conclusions?

Reviewer #1: Yes

Reviewer #2: Yes

3. Has the statistical analysis been performed appropriately and rigorously? 

Reviewer #1: I Don't Know

Reviewer #2: No

4. Have the authors made all data underlying the findings in their manuscript fully available?

Reviewer #1: Yes

Reviewer #2: Yes

5. Is the manuscript presented in an intelligible fashion and written in standard English?

Reviewer #1: Yes

Reviewer #2: Yes

6. Review Comments to the Author

Reviewer #1: After revision, the manuscript has improved significantly. Nevertheless, not all comments were fully clarified and there are some deficiencies in the study. The authors already published several results of their studies on takins (Yan et al. 2016 & 2017, Zeng et al. 2001, reference no. 19-21, 25, 30, 31, 39, 48, 54), thus this manuscript describes the results on the daily and seasonal pattern of the movement rate of takins. In general, it does not contain any significant new findings, even the authors themselves name 5 published studies on the activity pattern of takin (reference nos. 24-28).

The authors' lack of criticism of their own measurements and analyses is striking: Although the quite considerable (systematic) GPS error has been determined, it receives no further attention in the analysis or discussion. The error of the temperature data was not determined at all (I estimate it to be even higher in percentage than the GPS error). The measurement interval of 2 hours is very rough, but despite all these shortcomings, the authors calculate their movement rates to centimeter accuracy (Table 1) and estimate ecological effects without including calculated measurement errors. That is not state-of-the-art.

I have my doubts whether the statistical investigations are really scientifically correct: NDVI, temperature and season certainly correlate strongly, has this been taken into account? Why sometimes the maximum and sometimes the average movement rates were used, could no uniform value be found? The sample sizes from which the test data were calculated differ in the individual categories: male n = 4, female n = 6, different season durations (line 70-72) and day period durations (day, night, dusk, dawn), different total number of values per individual (from M3 = 2348 data = 53.6% of the maximum possible data in the measurement period, up to F6 = 4051 data = 92.5%). Systematic errors (GPS errors) of considerable size (in relation to the calculated movement data) were not taken into account in the analyses. Therefore, the variance within the data of a category can be much greater, which would definitely have an impact on the statistical results.

- Line 7: Delete "would

- Line 110: In which habitat and how exactly this GPS error was recorded? Is this value an average value from several measurements? What are the maximum values of the GPS error and how were outliers (maximum GPS errors) detected or filtered out of the GPS measurement data. Why was this GPS error not considered in the further analyses? On the one hand this GPS error seems to be very small for a typical Takin habitat (dense mountain forest). On the other hand, this error (about 11m) is very high compared to the calculated movement rates (on average about 60m/h or never more than 120m/h), so that an inclusion of this GPS error in the analyses would certainly be relevant for the results.

- Line 111-119: I think the sampling rate (12 values per day) is too low. The method´s reference study of Ensing et al. 2014 had a double sampling rate (24 values per day) and the authors themselves mention that the sampling rate has a decisive influence on the results (line 116/117). It is not known what the animals really did between the single measurements: a movement rate of zero m/hour could be based by different behaviors of the animal like lying or feeding at the same spot, running in circles, moving within the GPS error distance and so on. The greater this time span between the measurements, the greater the nescience. I already mentioned this in the first review, but the authors ignored this objection.

- Line 123-126: with this definitions, each dusk or dawn phase takes exactly 1.5 hours which is not in reality. Why was this division made this way? To get at least one GPS-location in each twilight phase? Estimating the effects, I would not have chosen the categories "dusk" and "dawn", because these twilight periods are too short for the sampling rate of 2 hrs. Instead, it would have been more useful to concentrate on the comparison "day" to "night" and hide the measurement around the twilight period. The activity peak in the twilight periods can be seen well in figure 2 anyway, the reason for using the categories "dawn" and "dusk" in the statistical tests was not explained and is not understandable to me.

- Line 127-129: Were 2 steps (one ends during the twilight time, the other begins during the twilight time) added to the respective twilight in this way? If so, the movement rates calculated from a total of 4 measurement intervals (in a total of 8 hours!) were attributed to the twilight phases; this is much longer than the duration of dusk and dawn in reality. In addition, the sample size per day segment differs: dusk/dawn with n = 1 each, "day" with n = 3-5, "night" with n = 5-7. Did you consider this misbalance in your analyses?

- Line 144-146: The data and test results for this correlation calculation are missing. Our own investigations with collar temperature data on animals in an enclosure and 2 temperature measurement stations (one in the shade, one in the sun) in this enclosure have shown that although these data correlate in the course of the day and year, there often occur considerable deviations of >10 degrees from the ambient temperature (e.g. if the outside temperatures exceed or fall below certain extreme values, if the collared animal takes a bath, if it moves and heats up or if the collar heats up in direct sunlight). The temperature collar data are therefore only suitable for rough analyses and comparisons (e.g. of daily averages). For using the collar temperature data on a smaller scale (like in this study), it is essential that a corresponding error interval is specified. This error in the temperature measurements may be quite high - similar to the GPS error in this study - and thus will also be relevant for the results. I guess it is well known that if 2 parameters "correlate significantly positive", this still does not mean that these two parameters are identical! Since in this study the analyses were made with the temperature data of the collars, these should not be called "ambient temperature" data (see line 147-149) just because they correlate with them!

- Line 152-156: The use of the parameters used in the statistical analyses (maximum or average movement rate) was not justified and is incomprehensible to me.

- Line 201-204, 252-254: How can there be this negative correlation between temperature and movement rate when the movement rate is always higher during the warm season or day period (summer, daytime) than during the cold ones (winter, night)?

- Line 234: "crepuscular" and "bimodal" mean the same thing.

- Line 224-226 & line 314-316: Isn't the seasonal change closely correlated with the change in temperature and NDVI? In this respect, you have simply noticed seasonally varying rates of movement, right? Since the length of day also correlates with seasonal change, I guess you would also find significant correlations with the movement rate if you used this parameter in the analyses.

- Line 279: "strategy" is misspelled.

- Discussion (line 237-240, line 256-259, line 271-274...): These statements about the behavior of the animals (midnight activity peak, sunbathing, feeding, rumination) or the exact time of certain activity peaks cannot be proven on the basis of the measured data and are therefore rather speculative. For such statements, either acceleration data (behavioral recording) must be measured and/or the sampling rate must be increased.

- Table 1: For what reason were these 6 parameters (in the first column) chosen? Which sample sizes are behind the individual values (these are certainly different sizes)? The test results are presented too confusingly.

- Figure 1: I would mention in the description that there were 6 females and 4 males.

- Figure 2: Is the unit (m/h) correct, because the data itself were collected every 2 hours?

- Figure S1: In the manuscript, the description or reference to the method of age estimation of the takins is missing. In the table heading "mass" should be replaced by "body mass"

Reviewer #2: This is a revised version of a manuscript on the locomotor activity patterns of takin. Although it is a smaller study with basic hypotheses, the deployment of GPS tracking, as a first for this species, and the correlation of space use with abiotic factors such as NDVI and temperature give the study value and make it a good contribution to its field.

The authors made a good effort to thoroughly revise the manuscript. Especially the language improved a lot. I can also see an effort to improve the statistical analysis, but still have some major concerns with it:

Although adopting a GLMM approach accounts for variation between individuals using the same model structure as before does not fix your problem with overfitting the model! You still have four factors in a model with a sample size of ten. Furthermore with calculating a GLMM you also add another factor as a random factor, further increasing the overfitting of the model. I do not think that your models are reliable at all. You have to reduce the number of fixed factors in your models. Additionally, there are not enough information on the models you ran (random factor? Distribution family? Link function?).

I have some minor concerns which are listed in the following:

Line 8: Change „difference“ to „differences“

Line 10-11: This sentence is still not correct. It is not a complete sentence. If you say “on the contrary during night time” you can not let the sentence just end, the sentence needs to include also what is different during night time. If you want the half sentence concerned with the night time to express, that during night time the movement rate of females was not significantly higher than that of males you can use “compare”: “In spring, the female movement rate was significantly higher than that of males during daytime compared to the movement rate during nighttime” or “In spring, the female movement rate was significantly higher than that of males during daytime but not during nighttime”

Or if during night time the movement rate of males was higher than that of females use “while”: “In spring, the female movement rate was significantly higher than that of males during daytime while during nighttime the movement rate of males was higher than that of females”

Line 46: Change “active” to activity

Line 68: Was there no weather data available during your study period?

Line 108-109: Please state in the manuscript why you chose this specific time frame for your study.

Line 139: change “Arcgis” to “ArcGIS”

Line 161-162: Die you statistically analyze whether there are seasonal differences in the time interval between movement peaks? If not you can not make a statement about whether or not it was longer during spring and summer compared to winter.

Line 177: Did you do post hoc testing to identify differences between groups? It is not clear were the differences between groups come from since the Friedman test should only provide a result for whether or not there is a difference between groups.

Line 205: please add a “the” before day

Line 206-207: It is enough to simply state “…was not correlated with ambient temperature.”

Line 230-231: You already stated in line 227 that you found a crepuscular activity pattern, you do not have to say it here again.

Line 287: change “significantly” to “significant”

Line 298: locomotor ability? Locomotor activity? Locomotor is an adjective and needs a noun in the sentence.

Line 304-305: You stated above that females mainly give birth and lactate during spring, here you are contradicting your previous statement. Which is it? Or do some females give birth in winter and others in spring?

7. PLOS authors have the option to publish the peer review history of their article (what does this mean?). If published, this will include your full peer review and any attached files.

Reviewer #1: No

Reviewer #2: No

---

## [Author Response · Author response to Decision Letter 1]

2 Jun 2020

Reviewer #1: After revision, the manuscript has improved significantly. Nevertheless, not all comments were fully clarified and there are some deficiencies in the study. The authors already published several results of their studies on takins (Yan et al. 2016 & 2017, Zeng et al. 2001, reference no. 19-21, 25, 30, 31, 39, 48, 54), thus this manuscript describes the results on the daily and seasonal pattern of the movement rate of takins. In general, it does not contain any significant new findings, even the authors themselves name 5 published studies on the activity pattern of takin (reference nos. 24-28).

The authors' lack of criticism of their own measurements and analyses is striking: Although the quite considerable (systematic) GPS error has been determined, it receives no further attention in the analysis or discussion. The error of the temperature data was not determined at all (I estimate it to be even higher in percentage than the GPS error). The measurement interval of 2 hours is very rough, but despite all these shortcomings, the authors calculate their movement rates to centimeter accuracy (Table 1) and estimate ecological effects without including calculated measurement errors. That is not state-of-the-art.

I have my doubts whether the statistical investigations are really scientifically correct: NDVI, temperature and season certainly correlate strongly, has this been taken into account? Why sometimes the maximum and sometimes the average movement rates were used, could no uniform value be found? The sample sizes from which the test data were calculated differ in the individual categories: male n = 4, female n = 6, different season durations (line 70-72) and day period durations (day, night, dusk, dawn), different total number of values per individual (from M3 = 2348 data = 53.6% of the maximum possible data in the measurement period, up to F6 = 4051 data = 92.5%). Systematic errors (GPS errors) of considerable size (in relation to the calculated movement data) were not taken into account in the analyses. Therefore, the variance within the data of a category can be much greater, which would definitely have an impact on the statistical results.

Line 7: Delete "would

OK. We accept this. 

- Line 110: In which habitat and how exactly this GPS error was recorded? Is this value an average value from several measurements? What are the maximum values of the GPS error and how were outliers (maximum GPS errors) detected or filtered out of the GPS measurement data. Why was this GPS error not considered in the further analyses? On the one hand this GPS error seems to be very small for a typical Takin habitat (dense mountain forest). On the other hand, this error (about 11m) is very high compared to the calculated movement rates (on average about 60m/h or never more than 120m/h), so that an inclusion of this GPS error in the analyses would certainly be relevant for the results.

OK. We accept this. We analyzed the effect of this GPS error for movement rate difference in our results.

- Line 111-119: I think the sampling rate (12 values per day) is too low. The method´s reference study of Ensing et al. 2014 had a double sampling rate (24 values per day) and the authors themselves mention that the sampling rate has a decisive influence on the results (line 116/117). It is not known what the animals really did between the single measurements: a movement rate of zero m/hour could be based by different behaviors of the animal like lying or feeding at the same spot, running in circles, moving within the GPS error distance and so on. The greater this time span between the measurements, the greater the nescience. I already mentioned this in the first review, but the authors ignored this objection.

OK. We accept this. Based on Zeng et al. (2001), takins would feed while walking. When lying at the same spot, takins had a rest or ruminated. Therefore, movement rate would indicate feeding activity of takins. 

Because of project capital limitation, the Lotak collars GPS7000 of we used were simple. One collar was scheduled to acquire the position every one hour in August 2013. However, this collar did not work before June 2014. Therefore, the collars of we used in this research were scheduled to acquire the position every two hours in June 2014. In these 10 collars, only 4 collars had worked in June 2016; other collars did not work because of battery running out before June 2016. 

- Line 123-126: with this definitions, each dusk or dawn phase takes exactly 1.5 hours which is not in reality. Why was this division made this way? To get at least one GPS-location in each twilight phase? Estimating the effects, I would not have chosen the categories "dusk" and "dawn", because these twilight periods are too short for the sampling rate of 2 hrs. Instead, it would have been more useful to concentrate on the comparison "day" to "night" and hide the measurement around the twilight period. The activity peak in the twilight periods can be seen well in figure 2 anyway, the reason for using the categories "dawn" and "dusk" in the statistical tests was not explained and is not understandable to me.

OK. We accept this. We did not choose the categories “dusk” and “dawn”. We would choose the categories “daytime” and “nighttime”. 

- Line 127-129: Were 2 steps (one ends during the twilight time, the other begins during the twilight time) added to the respective twilight in this way? If so, the movement rates calculated from a total of 4 measurement intervals (in a total of 8 hours!) were attributed to the twilight phases; this is much longer than the duration of dusk and dawn in reality. In addition, the sample size per day segment differs: dusk/dawn with n = 1 each, "day" with n = 3-5, "night" with n = 5-7. Did you consider this misbalance in your analyses?

OK. We accept this. We did not choose the categories “dusk” and “dawn”.

- Line 144-146: The data and test results for this correlation calculation are missing. Our own investigations with collar temperature data on animals in an enclosure and 2 temperature measurement stations (one in the shade, one in the sun) in this enclosure have shown that although these data correlate in the course of the day and year, there often occur considerable deviations of >10 degrees from the ambient temperature (e.g. if the outside temperatures exceed or fall below certain extreme values, if the collared animal takes a bath, if it moves and heats up or if the collar heats up in direct sunlight). The temperature collar data are therefore only suitable for rough analyses and comparisons (e.g. of daily averages). For using the collar temperature data on a smaller scale (like in this study), it is essential that a corresponding error interval is specified. This error in the temperature measurements may be quite high - similar to the GPS error in this study - and thus will also be relevant for the results. I guess it is well known that if 2 parameters "correlate significantly positive", this still does not mean that these two parameters are identical! Since in this study the analyses were made with the temperature data of the collars, these should not be called "ambient temperature" data (see line 147-149) just because they correlate with them!

OK. We accept this. The temperature data taken by collars were mostly affected by the solar radiation, shade and other factors of GPS location microenvironment, although it also affected by the heat radiation of the takins wearing the collar. Therefore, we called this temperature as “microenvironment temperature (MET)”. 

- Line 152-156: The use of the parameters used in the statistical analyses (maximum or average movement rate) was not justified and is incomprehensible to me.

OK. We accept this. We deleted this section.

- Line 201-204, 252-254: How can there be this negative correlation between temperature and movement rate when the movement rate is always higher during the warm season or day period (summer, daytime) than during the cold ones (winter, night)?

OK. In summer, temperatures are at their highest and takins move upward to forage in high altitude regions. Though temperatures in summer were highest during the year (Fig. 4), the takin movement rate was negatively correlated with MET during daytime (Table 1), indicating that the animals prefer to inhabit a cooler environment. The decrease in activity level in response to MET as a strategy against over-heating has previously been researched in other ungulates. During nighttime in winter, the takin movement rate was also negatively correlated with ambient temperature (Table 1), indicating that takins search for warm night habitats to avoid being exposed to the cold. Maintaining thermal balance is most important for the takins to select suitable habitats in winter.

- Line 234: "crepuscular" and "bimodal" mean the same thing.

OK. We accept this. We deleted “bimodal”.

- Line 224-226 & line 314-316: Isn't the seasonal change closely correlated with the change in temperature and NDVI? In this respect, you have simply noticed seasonally varying rates of movement, right? Since the length of day also correlates with seasonal change, I guess you would also find significant correlations with the movement rate if you used this parameter in the analyses.

OK. We accept this. We deleted these words.

- Line 279: "strategy" is misspelled.

OK. We accept this.

- Discussion (line 237-240, line 256-259, line 271-274...): These statements about the behavior of the animals (midnight activity peak, sunbathing, feeding, rumination) or the exact time of certain activity peaks cannot be proven on the basis of the measured data and are therefore rather speculative. For such statements, either acceleration data (behavioral recording) must be measured and/or the sampling rate must be increased.

OK. We deleted this section of Line 237-240. 

The takins prefer to lie down and bask long time in the sun in winter. This sunbathing of takins was often observed in winter based our field observation in many years, although we did not systematic analyzed this behavior. Therefore, we still retained this section.

We revised section of 271-274.

- Table 1: For what reason were these 6 parameters (in the first column) chosen? Which sample sizes are behind the individual values (these are certainly different sizes)? The test results are presented too confusingly.

OK. We accept this. We deleted this section.

- Figure 1: I would mention in the description that there were 6 females and 4 males.

OK. We accept this.

- Figure 2: Is the unit (m/h) correct, because the data itself were collected every 2 hours?

OK. We first calculated the Euclidean distance between successive locations (m). We subsequently calculated the true travel distance corrected using average change in altitude between successive locations. Finally, we calculated movement rate by dividing the true travel distance by the time elapsed between them (h).

- Figure S1: In the manuscript, the description or reference to the method of age estimation of the takins is missing. In the table heading "mass" should be replaced by "body mass"

OK. We accept this.

Reviewer #2: This is a revised version of a manuscript on the locomotor activity patterns of takin. Although it is a smaller study with basic hypotheses, the deployment of GPS tracking, as a first for this species, and the correlation of space use with abiotic factors such as NDVI and temperature give the study value and make it a good contribution to its field.

The authors made a good effort to thoroughly revise the manuscript. Especially the language improved a lot. I can also see an effort to improve the statistical analysis, but still have some major concerns with it:

Although adopting a GLMM approach accounts for variation between individuals using the same model structure as before does not fix your problem with overfitting the model! You still have four factors in a model with a sample size of ten. Furthermore with calculating a GLMM you also add another factor as a random factor, further increasing the overfitting of the model. I do not think that your models are reliable at all. You have to reduce the number of fixed factors in your models. Additionally, there are not enough information on the models you ran (random factor? Distribution family? Link function?).

OK. We accept this. We used one-way ANOVA to compare seasonal differences on movement rates during daytime, nighttime and whole day. We constructed models including the factors (sex, MET and NDVI) using the GLMM procedure in each season, random factor being animal individual.

I have some minor concerns which are listed in the following:

Line 8: Change „difference“ to „differences“

OK. We accept this.

Line 10-11: This sentence is still not correct. It is not a complete sentence. If you say “on the contrary during night time” you can not let the sentence just end, the sentence needs to include also what is different during night time. If you want the half sentence concerned with the night time to express, that during night time the movement rate of females was not significantly higher than that of males you can use “compare”: “In spring, the female movement rate was significantly higher than that of males during daytime compared to the movement rate during nighttime” or “In spring, the female movement rate was significantly higher than that of males during daytime but not during nighttime”

Or if during night time the movement rate of males was higher than that of females use “while”: “In spring, the female movement rate was significantly higher than that of males during daytime while during nighttime the movement rate of males was higher than that of females”

OK. We accept this.

Line 46: Change “active” to activity

OK. We accept this.

Line 68: Was there no weather data available during your study period?

OK. We accept this. We had acquired temperature data of the Foping weather station from 1981 to 2016.

Line 108-109: Please state in the manuscript why you chose this specific time frame for your study.

OK. We accept this.

Line 139: change “Arcgis” to “ArcGIS”

OK. We accept this.

Line 161-162: Die you statistically analyze whether there are seasonal differences in the time interval between movement peaks? If not you can not make a statement about whether or not it was longer during spring and summer compared to winter.

OK. We accept this. We deleted this section.

Line 177: Did you do post hoc testing to identify differences between groups? It is not clear were the differences between groups come from since the Friedman test should only provide a result for whether or not there is a difference between groups.

OK. We accept this. We deleted this section.

Line 205: please add a “the” before day

OK. We accept this.

Line 206-207: It is enough to simply state “…was not correlated with ambient temperature.”

OK. We accept this.

Line 230-231: You already stated in line 227 that you found a crepuscular activity pattern, you do not have to say it here again.

OK. We accept this. We deleted this section.

Line 287: change “significantly” to “significant”

OK. We accept this.

Line 298: locomotor ability? Locomotor activity? Locomotor is an adjective and needs a noun in the sentence.

OK. We accept this.

Line 304-305: You stated above that females mainly give birth and lactate during spring, here you are contradicting your previous statement. Which is it? Or do some females give birth in winter and others in spring?

OK. The birth season of takins occurred from February to April. Mother-infant relationship was very close when calf was infantile. In spring, calf moved with their mother. Mother gave care to her infant.

---

## [Decision Letter · Decision Letter 2]

17 Jun 2020

Locomotor activity patterns of takin (Budorcas taxicolor) in a temperate mountain region

PONE-D-19-21970R2

Dear Dr. Yan,

We’re pleased to inform you that your manuscript has been judged scientifically suitable for publication and will be formally accepted for publication once it meets all outstanding technical requirements.

Kind regards,

Julia Molnar, PhD

Academic Editor

PLOS ONE

Additional Editor Comments (optional):

Reviewers' comments:

Reviewer's Responses to Questions

**Comments to the Author**

1. If the authors have adequately addressed your comments raised in a previous round of review and you feel that this manuscript is now acceptable for publication, you may indicate that here to bypass the “Comments to the Author” section, enter your conflict of interest statement in the “Confidential to Editor” section, and submit your "Accept" recommendation.

Reviewer #2: All comments have been addressed

2. Is the manuscript technically sound, and do the data support the conclusions?

Reviewer #2: Yes

3. Has the statistical analysis been performed appropriately and rigorously? 

Reviewer #2: Yes

4. Have the authors made all data underlying the findings in their manuscript fully available?

Reviewer #2: Yes

5. Is the manuscript presented in an intelligible fashion and written in standard English?

Reviewer #2: Yes

6. Review Comments to the Author

Reviewer #2: This study about the locomotor activity patterns of takin and the factors shaping those patterns shows a clear sexual dimorphism in movement rates during the seasons and further provides evidence for a connection between takin movement rates and resource availability. Allthough the general research questions are not exactly new, the study is the first to employ GPS tracking to observe takin movement and therefore provides new and detailed insights into the behavioural adaptations of this species to ecological conditions, intrinsic requirements and life history, nicely complementing the allready existing studies.

The authors took large effort in adressing the comments raised before and addressed all concerns adequately. Especially the statisitical analysis was revised thoroughly and matches now the limitations of the data set.

I only have a few very small comments which should be easily fixed:

Line 36: change “would be” to ”were“

Line 199: Change “on” to “in” and delete the “in” after between

Line 342: There is an extra full stop at the end of the sentence ending with season

7. PLOS authors have the option to publish the peer review history of their article (what does this mean?). If published, this will include your full peer review and any attached files.

Reviewer #2: No

---

## [Editor Report · Acceptance letter]

1 Jul 2020

PONE-D-19-21970R2 

Locomotor activity patterns of takin (*Budorcas taxicolor*) in a temperate mountain region 

Dear Dr. Yan:

I'm pleased to inform you that your manuscript has been deemed suitable for publication in PLOS ONE. Congratulations! Your manuscript is now with our production department. 

Kind regards, 

on behalf of

Dr. Julia Molnar 

Academic Editor

PLOS ONE